# Uncovering Predictive Factors and Interventions for Restoring Microecological Diversity after Antibiotic Disturbance

**DOI:** 10.3390/nu15183925

**Published:** 2023-09-10

**Authors:** Jing Chen, Jinlin Zhu, Wenwei Lu, Hongchao Wang, Mingluo Pan, Peijun Tian, Jianxin Zhao, Hao Zhang, Wei Chen

**Affiliations:** 1State Key Laboratory of Food Science and Resources, Jiangnan University, Wuxi 214122, China; chen2021study@163.com (J.C.); luwenwei@jiangnan.edu.cn (W.L.); hcwang@jiangnan.edu.cn (H.W.); mingluopan@163.com (M.P.); pjtian@jiangnan.edu.cn (P.T.); jxzhao@jiangnan.edu.cn (J.Z.); zhanghao@jiangnan.edu.cn (H.Z.); weichen@jiangnan.edu.cn (W.C.); 2School of Food Science and Technology, Jiangnan University, Wuxi 214122, China; 3International Joint Research Laboratory for Pharmabiotics & Antibiotic Resistance, Jiangnan University, Wuxi 214122, China; 4(Yangzhou) Institute of Food Biotechnology, Jiangnan University, Yangzhou 225004, China; 5National Engineering Research Center for Functional Food, Jiangnan University, Wuxi 214122, China; 6Wuxi Translational Medicine Research Center and Jiangsu Translational Medicine Research Institute Wuxi Branch, Wuxi 214122, China

**Keywords:** gut microbiome, antibiotic disturbance, predictive recovery-associated bacterial species (p-RABs), ensemble learning framework, cross-cohort analysis, *Akkermansia muciniphila*

## Abstract

Antibiotic treatment can lead to a loss of diversity of gut microbiota and may adversely affect gut microbiota composition and host health. Previous studies have indicated that the recovery of gut microbes from antibiotic-induced disruption may be guided by specific microbial species. We expect to predict recovery or non-recovery using these crucial species or other indices after antibiotic treatment only when the gut microbiota changes. This study focused on this prediction problem using a novel ensemble learning framework to identify a set of common and reasonably predictive recovery-associated bacterial species (p-RABs), enabling us to predict the host microbiome recovery status under broad-spectrum antibiotic treatment. Our findings also propose other predictive indicators, suggesting that higher taxonomic and functional diversity may correlate with an increased likelihood of successful recovery. Furthermore, to explore the validity of p-RABs, we performed a metabolic support analysis and identified *Akkermansia muciniphila* and *Bacteroides uniformis* as potential key supporting species for reconstruction interventions. Experimental results from a C57BL/6J male mouse model demonstrated the effectiveness of p-RABs in facilitating intestinal microbial reconstitution. Thus, we proved the reliability of the new p-RABs and validated a practical intervention scheme for gut microbiota reconstruction under antibiotic disturbance.

## 1. Introduction

Many large numbers of microbes inhabit the human gut, and their symbiotic and mutualistic relationships with the host (interactions between host–bacteria and bacteria–bacteria) constitute a complex and dynamic ecosystem [1] which performs functions critical to the host, such as energy metabolism, immune homeostasis, and xenobiotic metabolism [2]. A growing number of studies have highlighted the pivotal role of commensal microbiota in human health, focusing largely on finding differences in the gut commensal microbiota of healthy adults [3] compared to patients suffering from diseases such as type 2 diabetes [4], obesity [5,6,7], inflammatory bowel disease [8,9], and colorectal cancer [10,11]. To this end, researchers have begun to explore the effects of external disturbances such as antibiotic exposure. Many studies suggest that antibiotic disturbance decreases microbial diversity [12,13,14], perturbs community structure and function, and may have long-term consequences such as antibiotic-associated diarrhea and resistance [15], slower development of the preterm infant microbiome [16,17], metabolic diseases in adults [18], and increased susceptibility to infection from opportunistic pathogens such as *Clostridioides difficile* [19].

Most studies have focused on the difference in intestinal flora caused by antibiotic diarrhea and *Clostridioides difficile* infection; however, we explored the effects of short-term broad-spectrum antibiotic treatment on the gut microbiome, as the recovery of gut microbes at this time may help prevent adverse outcomes such as diarrhea. Alpha diversity is widely used to quantify the extent of antibiotic-induced disruption and recovery level of gut microbes because of its ability to capture significant fluctuations in the community, thus serving as a direct indicator for distinguishing recoverers [20]. Particular species, such as recovery-associated bacterial species (RABs), can elucidate the differences between individuals and play a role in facilitating recovery following antibiotic treatment [21]. Although the one-sided Wilcoxon test in previous research has identified a set of RABs, uncertainty remains regarding their predictive capabilities in terms of accurately predicting outcomes. Moreover, it is not yet known whether additional predictive microecological indicators can be investigated for prognostic recovery after antibiotic disturbance.

Numerous studies have highlighted the potential association between high taxonomic and functional diversity or functional redundancy in the intestinal microbiome and recovery improvement after antibiotic disturbance [12,22,23,24,25]; however, specific evidence regarding their relationships remains limited. Investigating these relationships can provide a more comprehensive understanding of the complex dynamics underlying intestinal microbiome recovery and contribute to the development of predictive evaluation indices and targeted interventions to optimize recovery outcomes.

In this study, we obtained predictive recovery-associated bacterial species (p-RABs) that contribute to a more reliable assessment of gut microbiome recovery after antibiotic treatment. This was performed by conducting a cross-cohort analysis based on an ensemble feature learning diagram covering statistical tests, machine learning classifiers, and microbial association network analysis within a performance-driven framework. Subsequently, we proposed three predictive indices for gut microbiota recovery following antibiotic disturbance. Finally, by investigating the metabolic interactions among the identified p-RABs, we uncovered potential synergistic interactions that could be used as intervention strategies for gut microecological recovery. Validation experiments conducted in a mouse model demonstrated the effectiveness of the intervention in promoting the restoration of intestinal microecology.

## 2. Materials and Methods

### 2.1. Study Populations and Samples Selection

Canada (CA). Shotgun metagenomic datasets for a cohort of Canadian [12] individuals were obtained from the European Nucleotide Archive database (accession number PRJEB8094). This study analyzed stool samples obtained from healthy volunteers who received antibiotic treatment (cefprozil). Sampling was performed at three specific time points: before (day 0), during (day 7), and after antibiotic treatment (day 90).

Singapore (SG). The Singaporean cohort [21] consisted of individuals admitted to Tan Tock Seng Hospital in Singapore who were primarily prescribed co-amoxiclav and clarithromycin for a duration of 1–2 weeks. Notably, this cohort is not public. The stool samples were collected at three time points: pre-treatment, during treatment, and up to three months after antibiotic usage.

England and Sweden (EN and SW). The 16S rRNA sequencing datasets for the English and Swedish cohorts [26] were obtained from ENA (project ID: SRP057504). In both, healthy individuals were administered antibiotics (EN: amoxicillin; SW: clindamycin/ciprofloxacin). Stool samples were analyzed at three time points: day 0 (pre-treatment), day 7 (during treatment), and two months after antibiotic usage (post-treatment). These cohorts were used as training sets.

Denmark (DK1 and DK2, validation cohort). Shotgun metagenomic datasets for the two Danish cohorts were collected by the Danish Medical Authority and Gentofte University Hospital Hellerup (accession numbers: PRJNA588313 and ERP022986). In the DK1 cohort [27], healthy volunteers were administered ciprofloxacin, cefuroxime, doxycycline, or azithromycin for five days, and their stool samples were analyzed for pre-antibiotics (15 days before treatment), during treatment (day 5 at the end of the treatment), and post-treatment (day 30, one month after treatment). In the DK2 cohort [14], healthy volunteers were provided four days of treatment (vancomycin, gentamicin, and meropenem), and stool samples were analyzed on day 0 (pre-treatment), day 4 (during treatment), and day 42 (post-antibiotic).

America (US, validation cohort). The American cohort was a shotgun metagenomic dataset collected by Washington University in St. Louis School of Medicine (accession number: PRJNA664754). The cohort was administered vancomycin, gentamicin, and meropenem for five days. Stool samples were analyzed on day 0 (one day before antibiotics), day 6 (just after antibiotics, as during the treatment time point), and day 35 (post-treatment).

For all the above cohorts, all antibiotic treatment stage samples were further analyzed to identify RAB taxa and the samples between the training and validation datasets were selected at three sampling time points for further analysis.

### 2.2. Species-Level Taxonomic Profiling and Classification of Samples for All Cohorts

For metagenomic sequencing datasets, raw reads were quality-filtered and trimmed using the default options provided by famas (available at https://github.com/andreas-wilm/famas, accessed on 15 February 2023). To eliminate reads that potentially originated from the host genome, we mapped the hg19 by BWA-MEM (v0.7.17) [28] with default parameters. The remaining reads were utilized for taxonomic profiling using MetaPhlAn (v2.0) [29] with default parameters. The 16S rRNA sequencing datasets were mapped to the SILVA database [30] (v123) using Qiime 2 (v2022.2.1). Each read was matched to a species based on criteria such as identity (>97%) and query coverage (>95%). Finally, the assigned reads for each taxon were aggregated to generate an Operational Taxonomic Unit (OTU) table, as shown in Appendix A.

To facilitate cohort-specific association analysis and the identification of p-RABs, individuals within each cohort were categorized as either “recoverers” or “non-recoverers”. Referring to previous research that defined recoverers using Simpson’s diversity index [21], we calculated the gut microbial diversity based on the species level. Samples at post-treatment recovery to the interquartile range from the pre- and during-antibiotic treatments were defined as recoverers.

### 2.3. An Ensemble Learning Framework Used to Obtain p-RABs

Here, we proposed an ensemble learning framework to explore bacterial species that can predict microbiome recovery. The framework was composed of the following four parts:

PC-corr. We used discriminant principal component analysis [31] and set default parameters to obtain different combinations of potential features. Four PC-corr schemes were selected with the first three obtaining 23, 21, and 12 features when the cutoff values were set to 0, 0.1, and 0.2, respectively. The fourth scheme was to select 13 features from 23 (cutoff = 0) using a one-sided Wilcoxon test.

Netshift. The OTU tables were preprocessed and feature abundances >0.3 were selected for further experiments. Subsequently, the Spearman correlations between features were calculated and relationships with correlations >0.5 were retained. The resulting two files were entered into the Netshift website (https://web.rniapps.net/netshift, accessed on 30 September 2022) to obtain the microbial interaction network as well as the NESH (high scores represent greater variation) and DelBet (<0 represents driving disease to health) for each feature, and the final microbial species were identified by combining the two metrics.

Netmoss. The advantage of Netmoss [32] is that it can process batch effects and obtain healthy and unhealthy networks using the Netmoss score for the driving target. We plotted the precision–recall curve using the corresponding TPR and FPR values. The horizontal coordinate of the curve intersection is the threshold value of the Netmoss score; features greater than this value promote a state change. Finally, the directionality of the feature contribution is determined in combination with an interpretable SHAP package (v0.41.0).

Classifier. Here, Random Forest from scikit-learn (v1.0.2), XGBoost (v1.5.0), and LightGBM (v3.3.2) were used to select important features. First, three feature combinations were obtained from the three classifiers with the highest AUC. Then, we observed contributions of these features by SHAP analysis, and only selected the features that contributed to “recoverers”. Finally, XGBoost and LightGBM were added to our framework for analysis because these two classifiers not only had fewer features but also had higher AUC values.

*p*-value. A previous study proposed 21 RABs using a one-sided Wilcoxon test [21]. To ensure robustness and generalizability, cross-cohort analysis was employed to identify species that exhibited statistical significance in a minimum of two of the four cohorts under investigation.

In conclusion, six models were merged into an ensemble learning framework to identify the microbial taxa that collectively provided the most accurate predictions by leveraging the strengths of each model. The final identified p-RABs had the highest AUC (Appendix A).

### 2.4. Calculation of TD, FD, and FR within Samples

First, we searched the HMP reference genomes in the IMG/M-HMP data mart to find representative strains (preferably standard strains ATCC, CCTA, etc.) corresponding to all species in the OTU table of the Canadian (CA) cohort, and the gene annotation results of each strain were downloaded to construct the whole GCN. For our study, the annotated genomes of all the strains were downloaded from the IMG/M data mart. Second, FD, TD, and FR were calculated using MATLAB (v2019a) code based on the OTU and GCN tables. The calculation equations are as follows [33]:$${FR}_{\alpha }={TD}_{\alpha }-{FD}_{\alpha }=\sum_{i=1}^{N}\sum_{j\neq i}^{N}p_{i}p_{j}-\sum_{i=1}^{N}\sum_{j\neq i}^{N}p_{i}p_{j}d_{ij}$$

TD is quantified using Gini–Simpson index, which reflects the likelihood of selecting two members from the local community belonging to different taxa. Conversely, FD is evaluated using Rao’s quadratic entropy, which measures the average functional dissimilarity between any two randomly chosen members within the local community. In this context, the functional distance *d_ij_* = *d_ji_* ∈ [0, 1] between taxon *i* and taxon *j* represents the weighted Jaccard distance, which is calculated based on the genomes of the respective taxa. The microbial composition or taxonomic profile was denoted as *p* in these calculations.

### 2.5. Metabolic Interaction Network of p-RABs

To investigate the metabolic interactions among p-RABs, genome-scale metabolic models were obtained from the AGORA database [34] (v1.03). These models provide a comprehensive representation of the metabolic capabilities of the organisms under study. The metabolic support index [35] was employed to quantify the extent of metabolic interactions between p-RABs and to measure the percentage of metabolic reactions within an organism that became feasible in the presence of another organism. Simulations were conducted under anoxic conditions with a high-fiber diet supplemented with mucin- and bile-acid-derived metabolites [21] to mimic the relevant physiological context. Species pairs exhibiting high values of the metabolic support index (top 25%) were visualized using Gephi [36] (v0.9.7).

### 2.6. Microbial Food Web

Here, we used an existing food web [37] and visualized it using the hierarchical layout of Cytoscape [38] (v3.9.1) with annotation of p-RABs. It utilized the data mining technique of “association rule mining” and applied many gut microbiome data from the MEDUSA database to obtain 266 species and 1166 directed association edges—species that are in the upper layers have many incoming edges at the end of the food web and receive more metabolic support from the bottom species.

### 2.7. A Mouse Model of Microbiome Recovery after Antibiotic Treatment

#### 2.7.1. Strain Preparation

Strains (*Akkermansia muciniphila* BAA-835; *Bacteroides uniformis* CCFM1231; *Bacteroides thetaiotaomicron* FBJ10K7; and *Bifidobacterium adolescentis* CCFM643) maintained at the Culture Collection of Food Microorganisms in Jiangnan University, Wuxi, China (CCFM) were revived in BHI (add heme chloride and vitamin k) or MRS (only *Bifidobacterium adolescentis*, add cysteine) mediums under anaerobic conditions at 37 °C. *Bacteroides thetaiotaomicron* and *Bifidobacterium adolescentis* as the positive control group. The identified strains were harvested by centrifugation at 8000× *g* for 10 min at 4 °C, supplemented with glycerol (30%), and stored in a −80 °C refrigerator before animal experiments. The number of viable bacterial cells in the suspension was 1 × 10^8^ colony forming units (CFUs) for each strain.

#### 2.7.2. Animals and Design

The mouse experimental protocol was performed in strict accordance with the recommendations of the Experimental Animal Welfare Ethics Committee and was approved by the Laboratory Animal Center, Jiangnan University. The eight-week-old C57BL/6J male mice were purchased from Vital River Experimental Animal Technology, Shanghai, China. The mice were individually gavaged with ampicillin sodium salt (Sigma Aldrich, Shanghai, China) dissolved in 1 × PBS for five days at a dose of 100 mg/kg [39] as 2.5 mg for mice with 25 g [21] under specific-pathogen-free (SPF) conditions. After antibiotic treatment, the mice were divided into several groups (four mice per cage; two cages for each group; six experimental batches including the negative control batch) and gavaged with 1 × 10^8^ CFUs of *Akkermansia muciniphila* BAA-835; 1 × 10^8^ CFUs of *Bacteroides uniformis* FTJS3E4; 1 × 10^8^ CFUs of *Akkermansia muciniphila* BAA-835 + 1 × 10^8^ CFUs of *Bacteroides uniformis* FTJS3E4; 1 × 10^8^ CFUs of *Bacteroides thetaiotaomicron* FBJ10K7 + 1 × 10^8^ CFUs of *Bifidobacterium adolescentis* CCFM643; or 1 × PBS. The mice were maintained in an SPF environment at 22 ± 1 °C and relative humidity of 55 ± 10% with a 12 h light–dark cycle. Water and maintenance fodder was provided, and mice were placed in white plastic cages with corn cob bedding that had been pre-sterilized by autoclaving. On the morning of gavage, strains stored at −80 °C were removed from glycerol (8000× *g* for 10 min at 4 °C), washed twice, and dissolved in 1 × PBS (CFU of strains diluted into 1 × 10^8^), placed in sterilized centrifuge tubes, and then wrapped in tin foil and sent through the delivery window to the animal facility barrier.

#### 2.7.3. Fecal Sample Collection and DNA Extraction

Fecal samples were collected from each mouse by isolation in blank cages at multiple time points, including before antibiotic treatment (day 0) and at various time intervals during and after antibiotic administration (days 6, 7, 9, 11, 13, 15, and 17). Bacterial genomic DNA was extracted from the fecal samples using a Fast DNA Spin Kit (M Biomedicals Ltd., LLC, Irvine, CA, USA) specifically designed for fecal samples. To ensure the purity of the PCR products, a QIAquick Gel Extraction Kit (Qiagen GmbH, Hilden, Germany) was used following the manufacturer’s instructions. This stringent extraction and purification process was aimed at obtaining high-quality DNA samples for subsequent analyses.

#### 2.7.4. Taxonomic Profiling

Generation and sequencing of libraries followed established protocols described in a previous study [40]. Taxonomic classification was performed by mapping the reads to the SILVA database [30] (v123) using Qiime 2 (v2022.2.1). Each read was assigned to the species that corresponded to the best hit. Subsequently, alpha and beta diversity measurements were calculated using the R package “vegan”, providing valuable insights into the within-sample and between-sample diversity patterns.

#### 2.7.5. Health Status Assessment

We developed an index to evaluate the recovered and non-recovered populations by referring to the methods described in previous articles [41]. The index is the most intuitive way to determine how close a person’s microbiome is to that of a recovered (or unrecovered) population as it quantifies the balance between microbes associated with recovery and those associated with non-recovery. First, based on our experimental mouse data, the microbial species we identified should have different frequencies in the recovered group than in the unrecovered group. Next, an index was developed to evaluate the microbiome recovery. Finally, we evaluated classification accuracy using established indicators (Appendix A).

#### 2.7.6. Gut Microbiota Co-Occurrence Network Construction and Characterization

We constructed gut microbial co-occurrence networks for the negative control, model, and AmBu group mice on days 0, 6, 7, 9, 11, 13, 15, and 17. With the gut microbe profiles for each group, we calculated the Spearman correlation coefficients between genera using the R bag “psych” (v2.3.3) and kept correlations r < −0.4 or r > 0.4 and *p* < 0.05. We then plotted the gut microbial co-occurrence networks using Gephi [36] (v0.9.7). To characterize the topological features of the gut microbe co-occurrence networks, we used the “igraph” package (v1.4.1) in R and analyzed the numbers of nodes and edges and the mean degree in each network. In the gut microbe co-occurrence networks, each node indicated a genus and each link indicated a relationship between genera. Furthermore, we defined bacteria as hub nodes when their degrees were at the top 20% of degrees in the network.

Network vulnerability evaluation. We evaluated the vulnerability of the gut microbe networks [42], which were written using Perl code (v5.26.1). This method calculated the maximal global efficacy decreasing ratio (*mEDR*) as the maximal vulnerability of networks. Before determining the *mEDR*, we first calculated the average efficacy of a network (*E_a_*), which indicates the transfer speed of information in the network. The *E_a_* was calculated using the following formula:$$E_{a}=\frac{1}{n(n-1)}\sum_{i\neq j}\frac{1}{d_{i,j}}$$
where *n* is the number of nodes in networks a and *d_i_*_,*j*_ is the number of edges in the shortest path between nodes *i* and *j*. We then removed each node individually from the network, evaluated the altered $E_{a}^{\prime }$ after each removal, and selected the maximal *EDR* as the *mEDR* using the following formula:$$mEDR=\max\left(\frac{E_{a}-E_{a}^{\prime }}{E_{a}}\right)$$

## 3. Result

### 3.1. Better Predictive Recovery-Associated Bacterial Species Obtained from Different Methods

Here, we used 298 stool samples—obtained from seven independently published studies—from 91 individuals from six countries on three continents—Singapore [21], Canada [12], England [26], Sweden [26], Denmark [14,27], and America [13]—in a range of age groups (18–81 years). The cohort was divided into training and validation sets (Methods). We defined “recoverers” or “non-recoverers” using the Simpson index (Methods) [21] as it could distinguish between the two groups well. In the validation data (Appendix A), the recovery of microbial diversity exhibited a U-shaped profile, with a significant decrease in diversity during antibiotic disturbance, but a recovery in diversity at the post-treatment time point [21], and the intestinal microecology between the recoverers and non-recoverers was significantly different (*p* < 0.001, Appendix A).

Subsequently, we employed a performance-driven ensemble learning approach to integrate the p-RABs obtained using different methods into a unified predictive model. These methods can be broadly classified into four aspects: (1) PC-corr [31,43], which allows us to identify hidden combinations of features that promote antibiotic recovery; (2) microbial network analysis (Netshift [44], Netmoss [32]), which identifies key microorganisms that play a driving role in healthy–unhealthy transitions by examining the network topology; (3) machine learning algorithms (XGBoost, LightGBM), which construct classifiers and select key species features according to the feature importance; (4) statistical tests that directly observe the differences in feature distributions between recoverers and non-recoverers to identify key microorganisms. Each method analyzed data from three time points (pre-antibiotic, during-antibiotic, and post-antibiotic) to identify the crucial species contributing to the recovery of the gut microbiota, which needed to be present in at least two cohorts. Each method yielded its own set of p-RABs, considered separate units within the ensemble learning framework. By combining p-RABs from multiple methods, we aimed to identify the microbial taxa that collectively provided the most accurate prediction by leveraging the strengths of each category. Through this integrative analysis, we sought to enhance the accuracy and robustness of our predictions of gut microbiome recovery in the context of antibiotic treatment.

In summary, 52 p-RABs were ultimately identified as correlated with gut microbiome recovery (Appendix A), with 9 species identified in XGBoost, 12 in LightGBM, 10 in Netmoss, 16 in Netshift, 21 in *p*-value [21], and 23 in PC-corr. In particular, the PC-corr model can set different thresholds based on the correlation between features to discover more possible feature combinations. Here, we used three thresholds (cut-off), 0, 0.1, and 0.2, to obtain three p-RABs (number = 23, 21, and 12). The intersection between these methods is illustrated in the Venn diagram (Figure 1a).

By comparing the contribution of the microbial species obtained by these methods to the gut microbiome recovery (Appendix A), we found that the microbial species (p-RABs = 21, Figure 1b) assembled from XGBoost, LightGBM, and PC-corr (13 p-RABs; cut-off = 0, filtered with the Wilcoxon test) can better contribute to gut microbiome recovery after antibiotic treatment (Figure 1c). Particularly, the newly identified species predicted recovery better than the previously obtained species at pre-antibiotic treatment (Figure 1d). The excellent performance of p-RABs was also validated using the new datasets (Appendix A), suggesting that the identified p-RABs can better predict post-antibiotic recovery.

### 3.2. Within-Sample Taxonomic Diversity, Functional Diversity, and f_prabs_ Predict Gut Microbiome Recovery under Antibiotic Disturbance

The human gut microbiome contains a vast array of taxa with diverse gene families [45], contributing to taxonomic diversity (TD) and functional diversity (FD), and the remarkable functional consistency across individuals suggests notable functional redundancy (FR) within the human gut microbiome [46,47]. Previous studies have proposed that robust FR [48] observed in the human microbiome underlies its stability and resilience in response to perturbations [46,49]. We quantified the FR, FD, and TD of the human microbiome within samples based on a gene content network (GCN) framework [33]. For each sample, we graphed the relationship between the fraction of p-RABs represented as *f_prabs_* and two key factors: (1) the duration in days following antibiotic treatment referred to as *t*_post_; and (2) the TD, FD, and FR of the gut microbiota. We set two fractions of p-RABs: the first was determined by calculating the actual number of p-RABs present within each sample and expressing it as a ratio of the total number of species observed (Figure 2a–c, the first-type *f_prabs_*) while the second was expressed as the relative abundance ratio of p-RABs to all species within each sample (Figure 2d–f, the second-type *f_prabs_*).

Multiple linear regressions of the F test showed a significant positive association between *f_prabs_* and FD or TD, but not FR, and a stronger positive association between *f_prabs_* and TD than between *f_prabs_* and FD. Accordingly, the positive correlation between the second-type *f_prabs_* and FD or TD is more significant (*p* = 0.0003). These results suggested that high TD and FD levels of the gut microbiota increase the relative number of p-RABs; that is, high TD and FD contribute to the recovery of the intestinal microbiota after antibiotic disturbance. Moreover, we performed a multiple linear regression of previous RABs [21], but the results were not conclusive (Appendix A), as the correlation was still positive, but not significant, which further confirms that our p-RABs are more reasonable. Briefly, our findings indicate that the TD and FD of the gut microbiome may act as reliable indicators when confronted with disruptions, such as antibiotic treatment.

A growing number of studies have focused on quantifying the degree of antibiotic perturbation—such as with resilience metrics—highlighting the extent to which the ecosystem recovers to its pre-disturbed state [50,51], while others have focused on the mechanisms that lead to antibiotic recovery [21,52,53]. However, few studies have suggested methods to predict the recovery of gut microbes during antibiotic treatment. Based on the above results, we found that high FD and TD are more likely to promote gut microbiome recovery after antibiotic treatment and that they have a positive correlation with *f_prabs_*. Therefore, we used logistic regression equations to predict the recovery of the gut microbiome during antibiotic treatment, with TD, FD, and *f_prabs_* as dependent variables. The equation is as follows:$$y=0.4571FD+0.7309TD+0.4465f_{rabs}-0.8204$$

Logistic regression analysis showed that high FD, TD, and *f_prabs_* promoted the recovery of gut microbes after antibiotic treatment. The accuracy of the model was satisfactory, with the mean AUC value of 0.73 (Appendix A), and the mean accuracy of 0.76 indicating the usefulness of the proposed indicators for predicting antibiotic recovery. Additionally, we performed a correlation analysis between FD, TD, and *f_prabs_* before antibiotic treatment and found that both the Pearson and Spearman correlations were significantly positive (Appendix A), consistent with the above results.

### 3.3. Akkermansia muciniphila Plays an Important Role in Gut Microbes Recovery after Antibiotic Disturbance

Restoration of natural ecosystems is often driven by ecological interactions [54,55]. To understand the predictive p-RABs’ interactions and evaluate the benefits of the co-culture of different species, we built a microbial metabolic support interactions network (Figure 3, more details in Appendix A) using the Mequest algorithm [35] based on a genome-scale metabolic model [34]. The network showed that blue nodes—such as *Bacteroides ovatus* and *Bacteroides uniformis*—were able to provide some degree of metabolic support to other green nodes, suggesting synergism between them. These blue nodes contain the species highlighted at the bottom of the food web (Appendix A) and are all *Bacteroides* spp. The food web from bottom to top represents a clear dependency among bacteria within the intestinal microbiome. Specifically, the existence of species B (top level) seems to depend on the existence of species A (bottom level), whereas the reverse relationship is not true.

In the microbial metabolic support interaction network (Figure 3), *Akkermansia muciniphila* received the most metabolic support from the other p-RABs, and *Bacteroides uniformis* supported *Akkermansia muciniphila* the most (highest MSI value; more details provided in Appendix A). Thus, selecting these two species for intervention may be a valid choice. Furthermore, we used the Shapley value to rank the importance of the 21 features (Figure 4). The blue and red nodes indicate that the feature decreased and increased the prediction effect of the model in the sample, respectively. If feature values increase, along with increasing SHAP values, the feature contributes to positive samples (recoverers) and almost all species reacted the same. *Akkermansia muciniphila* ranked first or second in either XGBoost or LightGBM, and *Bacteroides uniformis* was not far behind in the order of importance. Therefore, we chose to use *Akkermansia muciniphila* and *Bacteroides uniformis* as the key species for animal experimental validation.

### 3.4. Synergy between Akkermansia muciniphila and Bacteroides uniformis Contributes to Rapid Reconstruction of Mice Intestinal Microecology after Antibiotics

To investigate the synergistic effects of *Akkermansia muciniphila* and *Bacteroides uniformis* on intestinal microbial recovery after antibiotics, C57BL/6J healthy male mice were used for in vivo experiments. Apart from the negative control group, the mice were treated with antibiotics for five days, and then randomly divided into the following five groups to study the therapeutic effect: model group (1 × PBS); positive control group (a combination of *Bacteroides thetaiotaomicron* and *Bifidobacterium adolescentis*) [21]; Am group (*Akkermansia muciniphila*); Bu group (*Bacteroides uniformis*); and AmBu group (a combination of *Akkermansia muciniphila* and *Bacteroides uniformis*). Intestinal microbiome recovery was monitored using stool samples collected at eight time points over an 18-day period, and the microbiome was analyzed by 16S rRNA sequencing, with six mice per group (Figure 5a).

As expected, all treatment groups, except for the negative control group, exhibited a clear reduction in microbiome diversity after antibiotic treatment (Figure 5b,c), accompanied by obvious changes in the structure of their microbial communities (Figure 5d). From the first (day 7) to the fifth day (day 11) after the intervention, the AmBu group exhibited the fastest richness recovery (Figure 5b). The Shannon diversity index of the AmBu group increased significantly on the first day after the intervention compared to the model group, but the positive control group did not (day 7; Figure 5c). Furthermore, the gut microbiome levels of the AmBu group matched that of the negative control group after the second gavage (day 9; Figure 5e). Compared to the model group, the AmBu group had more overlap with the confidence intervals (95%) than the negative control group in the final three days (Figure 5f). Considered the changes in bacteria, the relative abundance of beneficial bacteria such as *g_Bifidobacterium*, *g_Bacteroides*, *g_Parabacteroides*, *g_Akkermansia* and *g_Enterorhabdus* that decreased after antibiotic treatment can recover with gavaged *Akkermansia muciniphila* and *Bacteroides uniformis*, and *g_Enterococcus* that significantly increased after antibiotic treatment will decrease when provided with the intervention (Appendix A). These results suggest that AmBu colonization is sufficient to restore intestinal microbiome diversity. Additionally, the combination of Am and Bu promoted health improvements after antibiotic interference (Appendix A).

To better understand the changes in the gut microbial network in the AmBu group, Spearman correlation networks were constructed at eight different time points (detailed in the Methods section; Figure 6a–c). The number of nodes, edges, and mean degrees in the AmBu networks increased more after antibiotic disturbance than those in the model networks (Figure 6d–f). We also found that the number of hub nodes (the top 20% degrees of nodes as hub nodes) in the model networks increased slowly and only equaled half the number of AmBu networks on day 9 (Figure 6g). Moreover, the vulnerability [42] (robustness; see Section 2) of AmBu networks was always lower than that of the model networks (Figure 6h). These results indicate that the AmBu microbial networks were more stable over time.

## 4. Discussion

The commensal gut microbiota plays a critical role in human health and is used to detect and predict diseases [56]. The application of machine learning algorithms allows us to identify more microorganisms with predictive capabilities and other predictors [57]. Here, we constructed a novel ensemble learning model framework for recovery after antibiotic disturbance, using 298 samples from seven cohorts. Our results showed a better predictive effect than previous studies, and we therefore propose a reliable intervention for gut microbiome recovery after antibiotic disturbance in a mouse model. Notably, our study is unique in that we integrated several different single models to promote prediction accuracy and explored the process of gut microbiome recovery from a new perspective.

The utilization of an ensemble learning framework is a potent approach for mitigating the prediction bias associated with single models [58,59]. Moreover, through the incorporation of a cross-cohort analysis that considers potential confounding effects within individual studies, we were able to identify dependable and consistent associations related to gut microbiome recovery, despite variations in cohort characteristics, such as antibiotic use. The reliable p-RABs identified in this study—using the ensemble learning framework built upon cross-cohort analysis—could accurately predict and contribute to the recovery of the gut microbiome under antibiotic disturbance. Our model has clinical applicability, as it could serve as a valuable tool for guiding targeted intervention strategies to alleviate the adverse effects of antibiotics on the gut microbiome. Furthermore, the analytical approaches employed in this study hold promise for training machine learning models to predict microbiome recovery in various other disease contexts.

Additionally, this study used microbial indices, including TD and FD, which can serve as predictors of gut microbiome recovery. Consistent with the understanding that the gut microbiome TD is reduced under antibiotic treatment [60], individuals with higher TD and FD recover more quickly. This is in contrast to two studies on fecal microbial transplantation (FMT), which indicated that a higher level of FR within the recipient microbiota before FMT can impede the successful implantation of donor microbiota and potentially diminish the effectiveness of FMT [33]. A higher TD means that more species may be retained and more nutrients more easily utilized for the growth of the entire gut microbiota [61], with a richer FD indicating increased metabolic activity. Therefore, we speculated that the combination of three microbial markers—FD, TD, and the relative abundance ratios of p-RAB (*f_prabs_*)—could effectively predict the recovery of the human gut microbiome under antibiotic treatment, which is also applicable to microbiome recovery in other contexts.

Restoration of gut ecosystems is a complex process determined by microbial interactions [21,62,63]. The effects of synergistic interactions and microbial cross-feeding have been observed in p-RABs, contributing to a better understanding of microbiome recovery. While it is acknowledged that there are differences between the gut microbiota of mice and human [64], these distinctions do not undermine our capacity to validate the robustness of our model. Remarkably, *Akkermansia muciniphila* and *Bacteroides uniformis*, which are prevalent microorganisms in human and mouse gut microbiota [65,66], promote intestinal microbiome recovery and are consistently more effective than the other groups. We speculated that this may be due to the characteristics of *Akkermansia muciniphila* and therefore simulated a model to understand this. Our results showed that the following changes occurred after the intestinal microbiome was disturbed by antibiotics (Figure 7a): (1) taxon diversity decreases [67,68,69]; (2) the viscous layer became thinner [68]; (3) some bacteria showed excessive growth [67,68]; and (4) immune cell activity was inhibited [67,69]. Interventions with *Akkermansia muciniphila* and *Bacteroides uniformis* (Figure 7b) showed the following: (I) p-RABs represented by *Akkermansia muciniphila* rapidly and successfully colonized the intestinal mucosal layer by exploiting their mucin-degrading capabilities. The p-RABs—experts in the breakdown of complex dietary source carbohydrates such as *Bacteroides uniformis*—support increased metabolic activity [70,71,72]. (II) Breakdowns not only provide metabolic support for *Akkermansia muciniphila* but also support the repopulation of other bacteria that cannot degrade mucins and complex diet-derived carbohydrates [21,63,73]. (III) As the microbial community repopulates, the production of SCFAs increases, providing energy to the colonic epithelial cells and promoting mucus secretion [74]. (IV) SCFAs activate the immune system [75,76]. (V) The epithelial cells of the colon produce more mucin, such as MUC2, under the influence of mucus-associated bacterium *Akkermansia muciniphila* [77,78,79], and the thickness of the mucous layer increases, leading to a positive feedback loop that drives a faster recovery of the intestinal microecology [80]. In conclusion, *Akkermansia muciniphila* plays a pivotal role in promoting the turnover of the intestinal mucus layer for intricate regulation and balance, including mucus synthesis, secretion, and degradation, essential for maintaining the optimal protective function of mucus.

## 5. Conclusions

In general, the predictive factors that enhance the accuracy and robustness of predictions regarding gut microbiome recovery in the context of antibiotic treatment were identified here, as was the feasibility of *Akkermansia muciniphila* and complex-carbohydrate-degrading bacteria, such as *Bacteroides uniformis*, to restore the intestinal microecology. Our current work highlights the importance of *Akkermansia muciniphila* in promoting intestinal microbiome reconstruction and maintaining intestinal homeostasis after a short broad-spectrum antibiotic treatment. These results provide a better understanding of microbiome recovery. In future studies, we will focus on the function of p-RABs and how to guide personalized recovery through the dietary regulation of individual gut microbes during antibiotic treatment.

## Figures and Tables

**Figure 1 nutrients-15-03925-f001:**
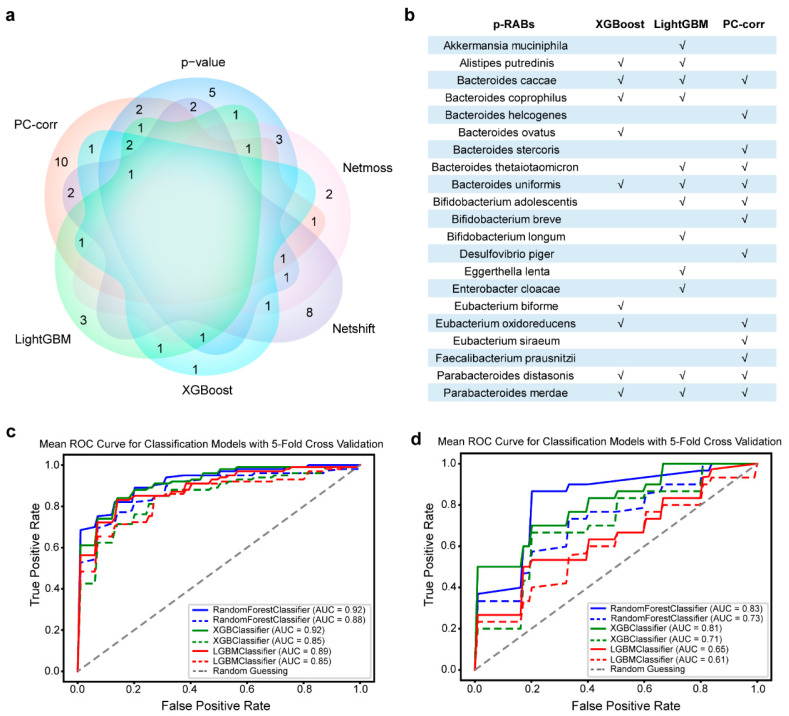
Final p-RABs identified through six different methods by ensemble thinking. (**a**) Venn diagram showing the number of microbial species obtained by six methods that can contribute to microbiome recovery. (**b**) The 21 microbial species (p-RABs) defined from the ensemble of XGBoost, LightGBM, and PC-corr (cut-off = 0 combined with one-side Wilcoxon test) can best predict intestinal microecological recovery after antibiotics. (**c**) The ROC curves and areas under the curves (AUC) for three models of the p-RABs (solid line) and the RABs (dashed line) pre-, during, and after antibiotic disturbance. (**d**) The prediction performance of the two sets of features pre-antibiotic disturbance.

**Figure 2 nutrients-15-03925-f002:**
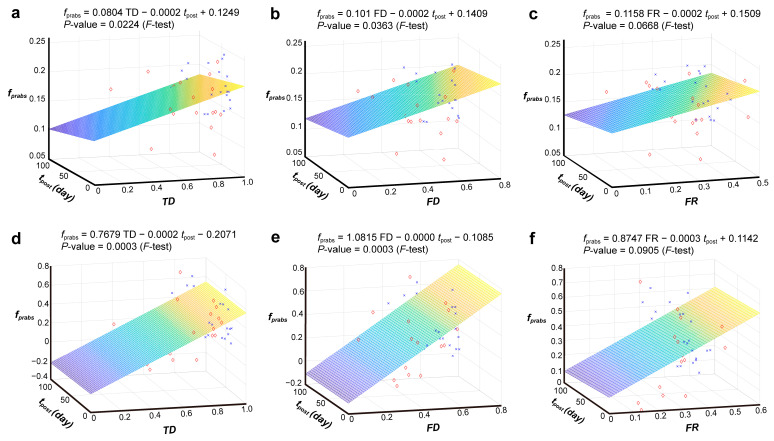
Multiple linear regression showed the effect of taxonomic diversity (TD), functional diversity (FD), and functional redundancy (FR) on recovery after antibiotic disturbance. (**a**,**d**) The taxonomic diversity (TD) from the Gini–Simpson index; (**b**,**e**) Rao’s quadratic entropy calculated as functional diversity (FD); and (**c**,**f**) the functional redundancy (FR) of gut microbiota, associated with days after antibiotic treatment and the two types of the fraction of p-RABs: (**a**–**c**) the number of present p-RABs; (**d**–**f**) the relative abundance ratio of p-RABs. Blue nodes represent recoverers and red nodes represent non-recoverers. These *p* values were calculated from the F test.

**Figure 3 nutrients-15-03925-f003:**
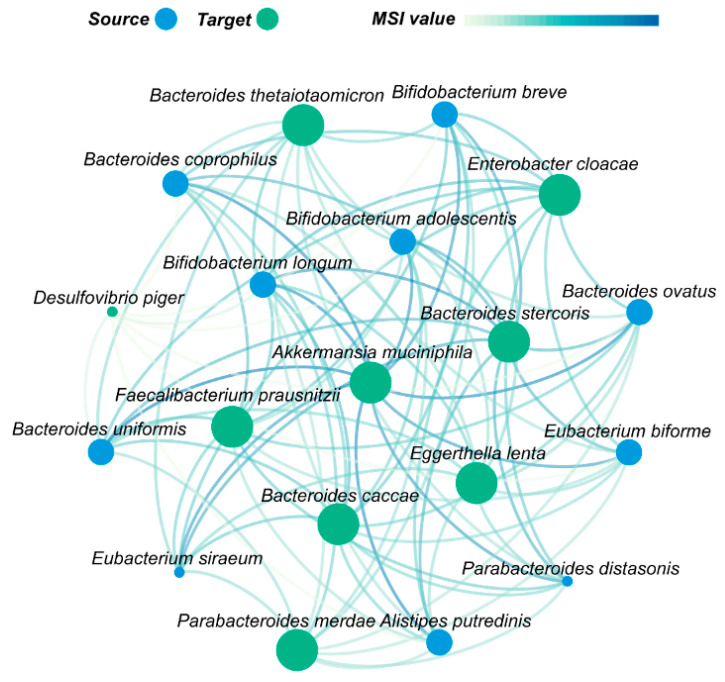
Crucial metabolic interactions between p-RABs. The node size reflects the number of connected edges; blue nodes indicate metabolic support for green nodes, the counterclockwise direction represents the direction of metabolic support, and the color of the line indicates the size of the MSI value, with darker colors indicating stronger metabolic support. The figure shows the p-RAB species with high metabolic support (MSI value in the top 25%).

**Figure 4 nutrients-15-03925-f004:**
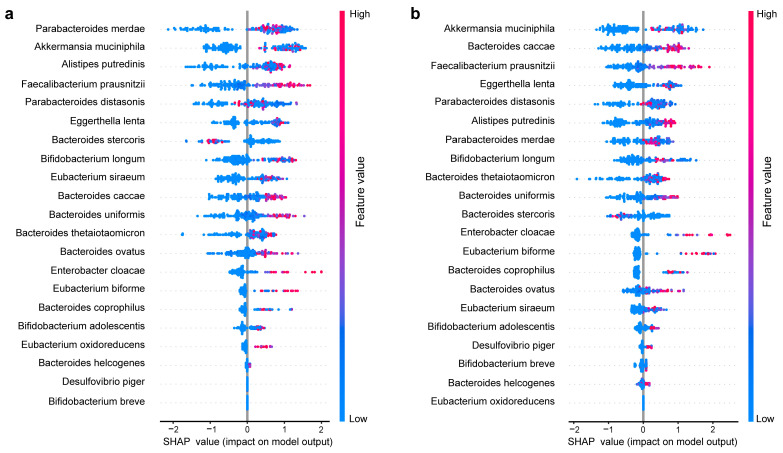
Ranking of feature importance of p-RABs based on SHAP package. (**a**) The feature importance ranking of the 21 p-RABs by the XGBoost classifier. (**b**) The feature importance ranking of the 21 p-RABs by the LightGBM classifier. The node color indicates the feature value, red to blue is the direction of the feature value decrease, and the SHAP value indicates whether the feature has a positive, negative, or no effect on the prediction.

**Figure 5 nutrients-15-03925-f005:**
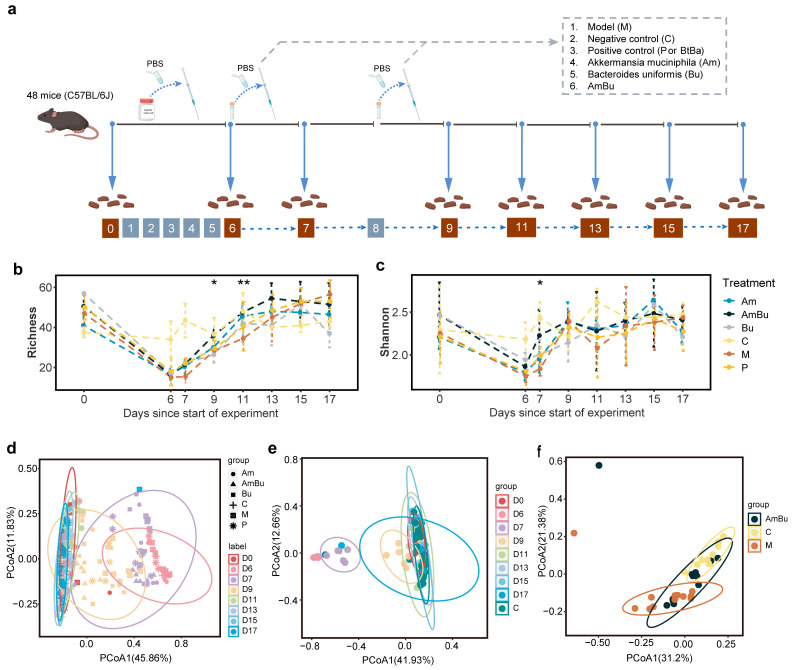
Promoting recovery of gut microbiome diversity in a mouse model using *Akkermansia muciniphila* and *Bacteroides uniformis*. (**a**) The schematic diagram describes the experimental process in a mouse model to investigate the effects of different species gavage groups on promoting intestinal microbiome recovery. Brown nodes indicate fecal sampling points; samples were taken every two days after the intervention (day 6 and day 8). (**b**,**c**) Observed species (richness) and microbiome diversity (Shannon) in six mouse groups across time. * *p* < 0.05 and ** *p* < 0.01 (one-sided Wilcoxon test) means the AmBu group significantly differs from the model group. (**d**) The difference between beta diversity calculated by Bray–Curtis distance in six different groups across time, eight colors for different time points, and six shapes for six groups. (**e**) The changes in mice microbiome composition across eight different time points of the AmBu group and compared with the negative control group, nine colors for different time points, and negative control samples. (**f**) The Bray–Curtis distance of gut microbiome between the negative control group, model group, and the AmBu group at day 15 and day 17 (the last two time points).

**Figure 6 nutrients-15-03925-f006:**
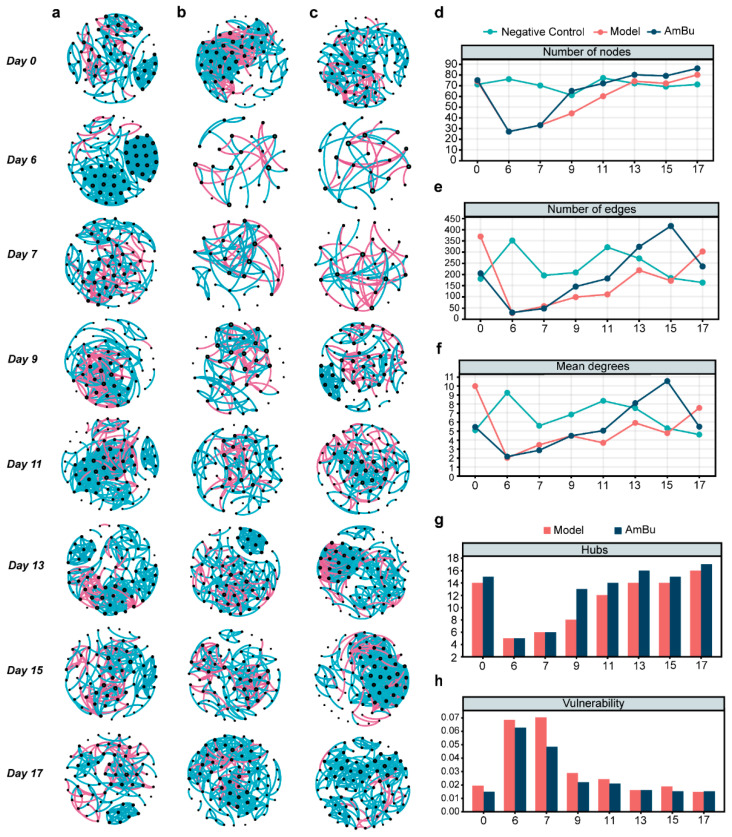
Dynamic changes in the gut microbiota co-occurrence networks and topological properties. The dynamic changes in the gut microbiota co-occurrence networks in (**a**) negative control, (**b**) model, and (**c**) AmBu group mice across eight time points. The number of (**d**) nodes, (**e**) edges, (**f**) mean degrees, and (**g**) hub nodes changes across different time points in the negative control, model, and the AmBu group. (**h**) The vulnerability of co-occurrence networks between model and the AmBu groups across eight time points.

**Figure 7 nutrients-15-03925-f007:**
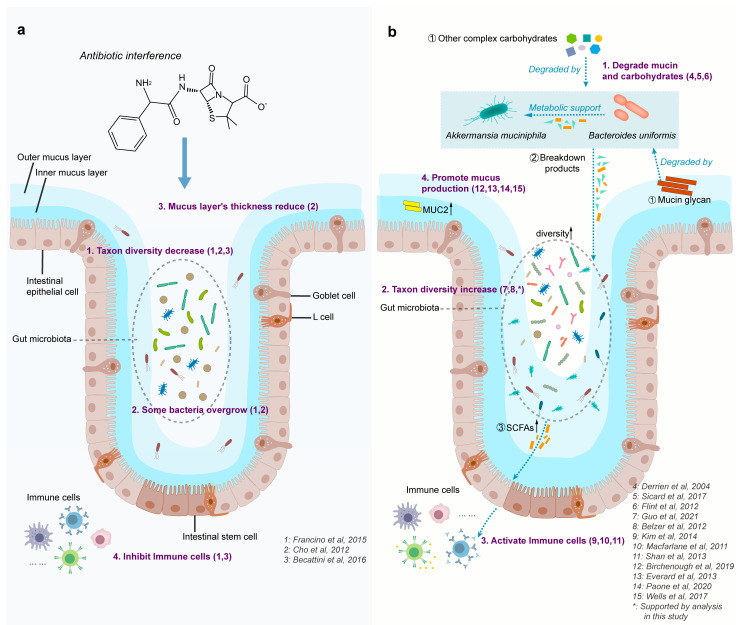
A model for gut microbiome based on observations and prior knowledge after antibiotic disturbance. (**a**) The left diagram simulates the effect of antibiotic disturbance on the gut, which is the premise of our intervention; the changes in the intestinal environment are shown in four ways [67,68,69]. (**b**) On the right, Am and Bu bacteria were provided by intragastric administration. There are three forms of carbon source transformation in the figure. Am and Bu can better colonize epithelial mucous membranes because they degrade mucus and dietary plant-, animal-derived carbohydrates (step 1) [70,71,72], then they break down carbohydrates and promote the growth of other species (step 2) [63,73]. Am, Bu, and the gut microbiota can produce more short-chain fatty acids (SCFAs) for the growth of colonocytes [74,77,78,79], which also activate immune cells (step 3) [75,76,80].

## Data Availability

The 16S rRNA sequencing data of the mouse gut microbiome were deposited in the NCBI Sequence Read Archive (SRA) repository under the BioProject (accession number PRJNA974222).

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
