# Peer review of "Uncovering Predictive Factors and Interventions for Restoring Microecological Diversity after Antibiotic Disturbance"

_nutrients, 2023, doi:10.3390/nu15183925_

Round 1

Reviewer 1 Report

A loss of diversity of gut microbiota after antibiotic treatment is an important problem.

The authors of this manuscript described that predictive recovery-associated bacterial species (p-RABs) plays an important role in microbiome recovery. I also agree with this theory.

The authors described that they identified two bacterial species, Akkermansia muciniphila and Bacteroides uniformis as a key factor supporting microbiome recovery. I have carefully read the data in this manuscript, but I think they do not fully prove the role of the two species. The authors should reconsider the role of the species in microbiome recovery. In addition, the authors should reconsider the role of bacterial species other than the two species in microbiome recovery.

The authors performed the experiments with human feces and that using a mouse model. However, there is a difference between human and mouse gut microbiota. They should describe the difference in the revised manuscript.

Author Response

Dear reviewer:

Reviewer 2 Report

The use of antibiotic therapy causes more and more the phenomenon of antibiotic resistance. But another rather emerging phenomenon is that of disturbing the microbiota ecosystem of the digestive system and in particular GUT microbiota. The proposed study therefore analyzes a rooted but emerging problem at the same time: in fact it determines the damage of the GUT microbiota after antibiotic therapy on pre-existing models. It is not limited to an observational study but implements a multi-centre evaluation involving some nutritional medicine centers such as Canada, Singapore, England, Sweden, Denmark, USA. Notably, this study is unique in that they integrated several models to promote prediction accuracy and explore the gut microbiome recovery process from a new perspective. Using an ensemble learning framework is a powerful approach to mitigating the prediction bias associated with individual models. Although conducted in mouse models, it has clinical applicability as it could serve as a valuable tool to guide targeted intervention strategies to alleviate the adverse effects of antibiotics on the gut microbiome. Finally, this study utilized microbial indices, including TD and FD, which may serve as predictors of gut microbiome recovery leading to preemptive findings: The effects of synergistic interactions and microbial cross-feeding observed in p-RABs, contributing to a better understanding of microbiome recovery. In a mouse model, Akkermansia muciniphila and Bacteroides uniformis promoted the recovery of the gut microbiome and were consistently more effective than the other groups.

Author Response

Dear reviewer:

Round 2

Reviewer 1 Report

I have reviewed the revised version.

The manuscript has been significantly improved.

I recommend that the manuscript be accepted for publication in Nutrients.